# A Standardized Botanical Composition Mitigated Acute Inflammatory Lung Injury and Reduced Mortality through Extracellular HMGB1 Reduction

**DOI:** 10.3390/molecules28186560

**Published:** 2023-09-11

**Authors:** Mesfin Yimam, Teresa Horm, Alexandria O’Neal, Ping Jiao, Mei Hong, Lidia Brownell, Qi Jia, Mosi Lin, Alex Gauthier, Jiaqi Wu, Kranti Venkat Mateti, Xiaojian Yang, Katelyn Dial, Sidorela Zefi, Lin L. Mantell

**Affiliations:** 1Unigen Inc., 2121 South State Street, Suite #400, Tacoma, WA 98405, USA; teresamarie522@gmail.com (T.H.); alexhoneal@gmail.com (A.O.); pjiao@unigen.net (P.J.); meih@unigen.net (M.H.); lbrownell@unigen.net (L.B.); qjia@unigen.net (Q.J.); 2College of Pharmacy and Health Sciences, St John’s University, Queens, NY 11439, USA; linm@stjohns.edu (M.L.); alexgauthier4@gmail.com (A.G.); wuj@stjohns.edu (J.W.); kranthi.mateti16@my.stjohns.edu (K.V.M.); xiaojing.yang@stjude.org (X.Y.); kd800@georgetown.edu (K.D.); zefi19@my.stjohns.edu (S.Z.); mantelll@stjohns.edu (L.L.M.)

**Keywords:** HMGB1, inflammatory lung injury, oxidative stress, sepsis

## Abstract

HMGB1 is a key late inflammatory mediator upregulated during air-pollution-induced oxidative stress. Extracellular HMGB1 accumulation in the airways and lungs plays a significant role in the pathogenesis of inflammatory lung injury. Decreasing extracellular HMBG1 levels may restore innate immune cell functions to protect the lungs from harmful injuries. Current therapies for air-pollution-induced respiratory problems are inadequate. Dietary antioxidants from natural sources could serve as a frontline defense against air-pollution-induced oxidative stress and lung damage. Here, a standardized botanical antioxidant composition from *Scutellaria baicalensis* and *Acacia catechu* was evaluated for its efficacy in attenuating acute inflammatory lung injury and sepsis. Murine models of disorders, including hyperoxia-exposed, bacterial-challenged acute lung injury, LPS-induced sepsis, and LPS-induced acute inflammatory lung injury models were utilized. The effect of the botanical composition on phagocytic activity and HMGB1 release was assessed using hyperoxia-stressed cultured macrophages. Analyses, such as hematoxylin-eosin (HE) staining for lung tissue damage evaluation, ELISA for inflammatory cytokines and chemokines, Western blot analysis for proteins, including extracellular HMGB1, and bacterial counts in the lungs and airways, were performed. Statistically significant decreases in mortality (50%), proinflammatory cytokines (TNF-α, IL-1β, IL-6) and chemokines (CINC-3) in serum and bronchoalveolar lavage fluid (BALF), and increased bacterial clearance from airways and lungs; reduced airway total protein, and decreased extracellular HMGB1 were observed in in vivo studies. A statistically significant 75.9% reduction in the level of extracellular HMGB1 and an increase in phagocytosis were observed in cultured macrophages. The compilations of data in this report strongly suggest that the botanical composition could be indicated for oxidative-stress-induced lung damage protection, possibly through attenuation of increased extracellular HMGB1 accumulation.

## 1. Introduction

The respiratory system is an easy target for constant endogenous and exogenous harmful attacks. Among the most pressing global challenges, inhalation of environmental fine particulate matter (PM2.5) from air pollution is a crucial risk factor that is known to cause significant oxidative and inflammatory damage to the lungs. Air pollution, including from wildfires, causes poor air quality problems and threatens the general public health. Recently, in 2022, researchers from the University of Southern California found a 5% increase in mortality risk on extreme PM2.5-only days compared with nonextreme days, highlighting the severe consequence of air pollution [1]. The mortality risk increased to 21% when heat was coupled with air pollution in this same survey. These recurrent exposures to air pollution are known to compromise the innate pulmonary immune response, leaving the lung susceptible to bacterial and viral infections as well as air-pollution-induced oxidative damage [2,3]. Older adults and people with weakened immune systems are more prone to and at greater risk of catastrophic health outcomes. Current conventional pharmaceutical intervention for this global health challenge is inadequate. Daily supplementation of antioxidants in the form of dietary polyphenols could potentially be considered as frontline defense and/or adjunct for air-pollution-induced oxidative stress damage of the lung.

Extracellular HMGB1 is among the key inflammatory mediators elevated by exposure to pollution. It is integral to oxidative-stress-associated downstream effects, surrendering the lung to injury and impeding its function [4,5]. It is a potent, late systemic inflammatory mediator known to serve as an alarmin to surrounding tissues, signaling the loss of cellular homeostasis and triggering a subsequent cytokine storm. The extracellular HMGB1 secreted passively by oxidative stressed cells or actively by activated innate immune cells is highly pro-inflammatory, with a crucial role in sepsis. High levels of HMGB1 in pulmonary tissues have been reported to be associated with a compromised immune response. An increased level of extracellular HMGB1 decreases alveolar macrophage phagocytic activity, subjecting sensitive lung tissue to pathogen invasion or pollution-related damage [6,7,8]. Antioxidants known to decrease the release of extracellular HMBG1 or inhibit its activity may restore the normal function of innate immune cells (i.e., alveolar macrophage phagocytic activity). For instance, the inhibition of HMGB1 expression in acute lung injury in murine models has been shown to reduce inflammation and tissue damage in addition to sustaining macrophage phagocytic activity, preserving immune activity and respiratory tract function [7,9,10,11,12].

Delivery of natural polyphenols in dietary supplements provides a unique advantage in oxidative stress management compared to the administration of simple antioxidant vitamins, as natural polyphenols have a greater structural diversity with a possibility of additional benefit than antioxidation [13]. In this regard, significantly diverse antioxidation properties have been reported for *Scutellaria baicalensis* and *Acacia catechu* extracts or their active constituents that make up the UP446 composition. For example, baicalin, a flavonoid from the root of *S. baicalensis*, has been found to have antioxidant properties that were significantly better than ascorbic acid and butylated hydroxytoluene (BHT) when evaluated on diphenylpicrylhydrazyl radical (DPPH) scavenging activity by iron-chelating assays [14]. It was also reported that baicalin and its aglycon, baicalein, scavenged hydroxyl radicals, DPPH radicals and alkyl radicals in a dose-dependent manner while effectively inhibiting lipid peroxidation of rat brain cortex mitochondria induced by ferrous ascorbic acid, AAPH or NADPH [15]. *S. baicalensis* root extract was also found to protect lipid peroxidation in lung tissue after free-radical-induced injury using linoleic acid hydroperoxide (LHP) [16]. Similarly, record numbers of health benefits of *A. catechu* and its active catechins have been reported through a reduction in ROS. For instance, *A. catechu* heartwood extract, characterized for its high catechin content, was assessed for its neuroprotection activity in both human neuroblastoma cells and rat brain tissues following hydrogen peroxide exposure. The extract was found to reduce ROS formation and protect the mitochondria from oxidative-stress-induced damage in both species [17]. Catechins exhibited strong properties of neutralizing reactive oxygen and nitrogen species in various oxidative-stress-induced lipid peroxidation assays in vivo and in vitro providing protection from oxidative-stress-induced damage [18,19].

UP446 is a standardized composition consisting primarily of a free B-ring flavonoid, baicalin, from *S. baicalensis* and a flavan, catechin, from the heartwoods of *A. catechu* as detailed in the materials and methods section. The composition is a dual cyclooxygenase (COX) and lipoxygenase (LOX) enzyme inhibitor which was found to decrease mRNA expression and protein levels of the proinflammatory cytokines, such as interleukin (IL)-1β, IL-6, and tumor necrosis factor (TNF)-α in preclinical studies [20,21]. Recently, it has been reported that the composition was considered beneficial in mounting a robust humoral response (elevated total IgA and IgG levels) in healthy participants following influenza vaccination paired with strong antioxidation capacity (increased glutathione peroxidase) in a randomized double-blind placebo control clinical trial [22]. These preclinical and clinically proven anti-inflammatory, antioxidant and immune support properties of the composition may have significant contributions to a healthy respiratory system. The current studies were designed to explore additional mechanisms by which the composition could provide protection to the lung and/or the respiratory system in general.

The botanicals constituting the composition tested in this report have been studied separately or together and indicated for various uses in respiratory system support. *S. baicalensis* was recorded in the classical Chinese medical literature (Shen Nong Ben Cao) from the Eastern Han dynasty (circa 200 C.E. or 2200 years ago). A recent list of the top 30 herbs in Traditional Chinese Medicine (TCM) for treating respiratory infections based on the analysis of two TCM databases (the World Traditional Medicine Patent database (WTM) and the Saphron TCM database) had *Radix Scutellariae* as the second most utilized herb, with a 38% frequency in all TCM compositions for treatment of respiratory tract infections [23].

Bioflavonoids, mainly baicalin and its aglycon, baicalein, have been identified as the bioactive components of *Radix Scutellariae*, with biological functions related to antioxidation, anti-inflammation, reduction in the allergic response, and antibacterial activity [24]. Broad spectrum anti-viral activity of these bioflavonoids for commonly isolated causative agents of respiratory tract infection has been reported from in vitro and in vivo studies. For instance, baicalin and baicalein exhibited potent antiviral activity through the inhibition of proteins that viruses need to bind to and bud from host cells–activities that are essential for infection [25]. In mice infected with Influenza A H1N1 virus (swine flu), an extract from *Radix Scutellariae* modulated their inflammatory response to reduce disease severity, decreased lung tissue damage, and ultimately increased their survival rate [26]. Recently, it was also found that the ethanol extract of *S. baicalensis* inhibits SARS-CoV-2 3CLpro in vitro with an IC_50_ of 8.52 µg/mL and inhibits the replication of SARS-CoV-2 virus in Vero cells with an EC_50_ of 0.74 μg/mL [27]. Among the major components of *S. baicalensis*, baicalein strongly inhibits SARS-CoV-2 3CLpro activity in vitro with an IC_50_ of 0.39 μM and inhibits replication of SARS-CoV-2 in Vero cells with an EC_50_ of 2.9 µM. The study demonstrates that the ethanol extract of *S. baicalensis* limited coronavirus replication in live cells at a concentration that could be achieved following oral administration for a clinically meaningful outcome [27].

Similarly, *Acacia catechu* has been used in ayurvedic medicine for centuries, with significant science-based reports suggesting its application in respiratory system support. *Acacia catechu* has been found to increase the number of antibody-producing cells, increase macrophage phagocytic activity, and inhibit the release of pro-inflammatory cytokines [28]; it has been found to be beneficial in ameliorating chemically induced oxidative stress, inflammation, and apoptosis in the lungs of mice [29], and it has been shown to increase immune modulation effects on both cell-mediated and humoral immunity in vivo [30].

Recently, Feng et al. have shown that a standardized blend of *Scutellaria baicalensis* and *Acacia catechu* may have a therapeutic advantage for COVID-19 through a “multi-compound and multi-target” approach to directly inhibit the virus, improve immune function, and reduce the inflammatory response associated with COVID-19. The author’s suggestion was based on a systems pharmacology methodology integrated by ADME screening, target prediction, network analysis, gene ontology (GO), Kyoto Encyclopedia of Genes and Genomes (KEGG) enrichment analysis, molecular docking, and molecular dynamic simulations [31].

Given these research-based findings for *S. baicalensis* and *A. catechu*, in the present study, we evaluated the effect of an antioxidant botanical composition (UP446) on the level of HMGB1 release from hyperoxia-stressed macrophages in vitro and assessed its impact on survival in a lipopolysaccharide (LPS)-induced sepsis and acute inflammatory lung injury model. We further expanded our evaluations and tested the composition of hyperoxia-exposed and bacteria (*Pseudomonas aeruginosa*)-challenged acute lung injury in vivo for its application in respiratory system support.

## 2. Results

### 2.1. Effect of UP446 on LPS-Induced Mortality Rate

Three hours following intraperitoneal injection of LPS, mice started to show early signs of endotoxemia. Exploratory behavior of mice was progressively reduced and was accompanied by ruffled fur (piloerection), decreased mobility, lethargy, and diarrhea. While these signs and symptoms seemed to be present in all the treatment groups, the magnitude of severity was more pronounced in the vehicle-treated group.

Two mice from the vehicle-treated group were found deceased 24 h after LPS injection. The survival rate was determined for this group and was found as 62.5% at this time point (Figure 1). Mice treated with UP446 had a 100% survival rate 24 h after LPS injection. Survival rates of 87.5% and 50% were observed for mice treated with UP446, and vehicle, respectively, 34 h after LPS injection. Perhaps the most significant observation for UP446-treated mice was observed 48 h after LPS injection. At this time point, there was only a 12.5% survival rate for the vehicle-treated mice, while UP446-treated mice showed a 75% survival rate. On the third day (72 h after LPS injection), the survival rates for the groups were 62.5% and 12.5% for UP446 and vehicle, respectively. All mice in the vehicle control group were deceased after 82 h of LPS injection, leaving a 0% survival rate for this group.

On the other hand, mice treated with UP446 showed a 50% survival rate and remained the same 96 h and 120 h after LPS injection. This survival rate was statistically significant (*p* = 0.001) when compared to the vehicle-treated animals (Figure 1). Surviving animals showed progressive improvements in their well-being. Mice appeared physically better and gradually resumed normal behaviors.

### 2.2. Effect of UP446 on LPS-Induced Acute Inflammatory Lung Injury

The severity of lung damage as a result of intratracheal LPS was assessed using H&E-stained lung tissue (Figure 2A). Rats in the vehicle-treated group showed statistically significant increases in the severity of pulmonary edema (2.5-fold increase) (Figure 2B) and lung damage (3.5-fold increase) (Figure 2C). Daily oral treatment of rats for a week with the high dose of UP446 at 250 mg/kg resulted in a statistically significant 20.8% reduction in overall lung damage severity when compared to vehicle-treated, LPS-induced, acute lung injury rats (Figure 2C). Similarly, a strong trend in the reduction in pulmonary edema (23.3% reduction, *p* = 0.08) was observed for the high dose of UP446 when compared to the vehicle-treated rats (Figure 2B). The low dose UP446 group saw minimal changes in the histopathology evaluation relative to the vehicle-treated diseased rats.

Statistically significant elevations in proinflammatory cytokine and inflammatory protein levels were observed for vehicle-treated rats challenged with LPS. These increases were significantly reduced when rats were treated with UP446 (Figure 3A–E). Statistically significant and dose-correlated reductions were observed for rats treated with UP446 at 250 mg/kg and 125 mg/kg orally. These reductions were calculated against the vehicle control and were found to be 90.7% and 69.8% for TNF-α (Figure 3A) and 81.2% and 61.8% for IL-1β (Figure 3B) when UP446 was administered at 250 mg/kg and 125 mg/kg, respectively. While the highest dose (250 mg/kg) resulted in a 74.6% reduction in the level of BAL IL-6, the lower dose showed a 58.3% reduction (Figure 3C). In this study, LPS rats treated with the high dose of UP446 showed statistically significant 42.4% reductions in CRP compared to the vehicle-treated disease model (Figure 3D).

The daily oral treatment of UP446 at 250 mg/kg for a week caused a statistically significant reduction in cytokine-induced neutrophil chemoattractant 3 (CINC-3) in LPS-induced acute lung injury animals (Figure 3E). The level of CINC-3 in the normal control rats receiving only the PBS intratracheally was near zero. In contrast, intratracheal LPS-induced acute lung injury rats treated with the carrier vehicle showed an average lung homogenate level of CINC-3 at 563.7 ± 172.9 pg/mL. This level was reduced to an average value of 360.8 ± 110.7 pg/mL for the 250 mg/kg UP446 treated rats. This 36% reduction in CINC-3 level for the rats treated with 250 mg/kg of UP446 was statistically significant when compared to the vehicle-treated disease model. The lower dose UP446 group only had a marginal 10.5% reduction in lung homogenate CINC-3 level in comparison to the vehicle-treated rats.

### 2.3. Effect of UP446 on Hyperoxia-Exposed, Bacteria-Challenged Mice

Pre-exposure to hyperoxia (O_2_) caused significantly more severe acute lung injury, indicated by the amount of protein secretion and lung edema in these mice compared to the mice that remained at room air (RA) (Figure 4A). The composition UP446 significantly reduced this effect by 58.1%. The reduction in the total protein content in lung lavage fluids of mice in the UP446-treated group was statistically significant compared to that of mice treated with hyperoxia and vehicle control (O_2_).

Bacterial loads in the airways (Figure 4B) and lung homogenates (Figure 4C) were elevated 36- and 44-fold, respectively, by preexposure of the mice to hyperoxia (O_2_), compared to those of mice that remained at room air (RA). Mice treated with UP446 had significantly lower (88.3% and 88.8% reductions) bacterial loads in their airways and lung homogenates, respectively, compared to mice exposed to hyperoxia and treated with vehicle alone. These differences in the bacterial loads in airways and lung homogenates were statistically significant compared to that of mice treated with hyperoxia and vehicle control (O_2_).

### 2.4. Effect of the Composition on HMGB1 Release

Mice with hyperoxia-induced lung injury that were challenged with *Pseudomonas aeruginosa* (PA) bacteria showed a 5-fold increase in extracellular HMGB1 in the lung lavage compared to the room air (RA) mice challenged with PA. Pretreating animals with UP446 showed a 71.6% reduction in the level of extracellular HMGB1 in the lung lavage compared to vehicle-treated mice exposed to hyperoxia and PA infection (Figure 5A). These reductions were statistically significant for both the positive control and UP446 groups.

Compared to the room air control group (21% O_2_) (RA), HMGB1 release in the hyperoxia control group (95% O_2_) (O_2_) was significantly increased. The vehicle, DMSO, did not significantly alter HMGB1 release compared to the hyperoxia control group. In contrast, treatment with the composition UP446 resulted in dose-correlated, statistically significant reductions (75.9–89.7%) in the level of HMGB1 when tested at 3.7 μg/mL, 11.1 μg/mL and 33.3 μg/mL (Figure 5B).

### 2.5. Effect of the Composition UP446 on Macrophage Phagocytic Activity

Cultured macrophages (RAW264.7) were subjected to hyperoxia for 24 h in the presence of either different concentrations of the composition UP446 or the vehicle alone. As indicated in the images, hyperoxia exposure significantly compromised macrophage phagocytic activity. The composition UP446 at doses as low as 3.7 µg/mL significantly enhanced macrophage function. Increasing the concentration further to 11.1 and 33.3 µg/mL did not seem to enhance the phagocytic activity of macrophages from what was already observed for the lowest concentration, though a slight increase was seen for the higher concentration tested. Optimum activation of macrophages for their phagocytic activity was already observed at the minimum concentration tested. The phagocytosis activity from each UP446 concentration was statistically significant (Figure 6). These results suggest that the composition UP446 enhances alveolar macrophage phagocytic activity, protecting lung functions under oxidative stress.

## 3. Materials and Methods

### 3.1. The Botanical Composition

A proprietary botanical composition UP446 is a mixture of standardized extracts from roots of *S. baicalensis* and heartwoods of *A. catechu* with a baicalin content not less than 60% and aa catechin content not less than 10% in the composition (Figure 7). Other minor flavonoids, such as wogonin 7-glucuronide and baicalein, etc., account for about 15% of the total weight. Moisture, ash, fat, and fiber constitute the remaining weight. A detailed method for the preparation of the two major flavonoids, baicalin and catechin, from the roots of *S. baicalensis* and the heartwoods of *A. catechu*, respectively, was disclosed in a US patent [32].

### 3.2. Lipopolysaccharide (LPS)-Induced Sepsis in Mice

Purpose-bred CD-1 mice purchased from Charles River Laboratories at the age of 8 weeks were used for this study. Mice were acclimated for a week before being assigned to study groups. Groups included: G1 = Normal control, G2 = vehicle control (0.5% carboxymethyl cellulose (CMC)), and G3 = UP446 (250 mg/kg). Eight mice were allocated to each group. Mice were pretreated with the composition (UP446) for 7 days before receiving a lethal dose intraperitoneal injection of LPS (*E. coli*, 055:B5; Sigma, St. Louis, MO, USA; Lot# 081275) at 25 mg/kg with a 10 mL/kg PBS volume. LPS was dissolved in phosphate-buffered saline (PBS; Lifeline, Lot # 07641). Animals were observed hourly and monitored for 5 days after LPS injection. The survival rate compared LPS + vehicle (0.5% CMC; Spectrum, New Brunswick, NJ, USA; Lot # 1IJ0127) and LPS + UP446. Mice were kept in a temperature-controlled room and were provided with food and water ad libitum.

### 3.3. Lipopolysaccharide (LPS)-Induced Acute Inflammatory Lung Injury in Rats

The study was designed to evaluate the direct impact of the composition UP446 in alleviating LPS-induced acute lung injury, with daily oral administration at 250 mg/kg (high dose) and 125 mg/kg (low dose). Animals were pretreated with the test materials for 7 days before model induction with LPS. On the 8th day, an hour after oral treatment, LPS was delivered intratracheally (i.t.) at 10 mg/kg, dissolved in 0.1 mL/100 g PBS, to each rat. The normal control rats received the same volume i.t. of PBS only. Groups included G1 = Normal control without LPS, G2 = vehicle control (0.5% CMC) with LPS, G3 = UP446 high dose (250 mg/kg) with LPS and G4 = UP446 low dose (125 mg/kg) with LPS. Ten rats were allocated to each group.

Surviving animals were sacrificed 24 h after intratracheal LPS administration. At necropsy, bronchoalveolar lavage (BAL) was collected by intratracheal injection of 1.5 mL PBS into the right lobe of the lung, followed by gentle aspiration at least 3 times. Recovered fluid was pooled, centrifuged at 1500 rpm for 10 min at 4 °C, and was used to measure cytokines (e.g., IL-6) and pulmonary protein level. This same right lobe was collected for tissue homogenization from each rat for MIP-2/CINC-3 protein quantification. The left lobe was fixed with formalin and submitted for histopathology evaluation to Nationwide Histology for analysis by a certified pathologist. Serum collected at necropsy was used to measure cytokines, such as TNF-α and IL-1β.

### 3.4. The Effects of the Botanical Composition on Hyperoxia-Exposed-, Bacterial-Challenged, Acute Inflammatory Lung Injury In Vivo

The effect of the botanical composition UP446 on bacterial clearance and HMGB1 expression level was evaluated in vivo. Male C57BL/6 mice (6 to 10 weeks old; The Jackson Laboratory, Bar Harbor, ME, USA) were used in this study, which was approved by the Institutional Animal Care and Use Committee at St. John’s University. The mice were housed in a pathogen-free environment, maintained at 22 °C and 50% relative humidity with a 12 h light/dark cycle. All mice had ad libitum access to standard rodent food and water. Mice were randomized to G1 = Normal control without bacterial inoculation, G2 = vehicle-treated disease model, and G3 = Disease model + UP446 (250 mg/kg). Following treatments with the vehicle, or the composition UP446 at 250 mg/kg orally for seven days, mice were exposed to >90% oxygen for 48 h and continued oral treatment for 2 more days before being inoculated with 0.1 × 10^8^ colony-forming units (CFUs) of *Pseudomonas aeruginosa* (PA). Mice were placed in microisolator cages (Allentown Caging Equipment, Allentown, NJ, USA) that were kept in a plexiglass chamber (Bio-Spherix, Lacona, NY, USA) for hyperoxia exposure. An oxygen analyzer (MSA; Ohio Medical Corporation, Gurnee, IL, USA) was used to monitor the O_2_ concentration in the chamber. Mice were euthanized 24 h after bacteria inoculation and bronchoalveolar lavage (BAL) fluid was collected. The lungs were gently lavaged twice with 1 mL of a sterile, nonpyrogenic phosphate-buffered saline (PBS) solution (Mediatech, Herndon, VA, USA), containing a cocktail of protease and phosphatase inhibitors (Pierce, ThermoFisher Scientific, Waltham, MA, USA). BAL samples were centrifuged at 200× *g* at 4 °C for 5 min, and the resultant supernatants were immediately used for quantitative bacteriology. Lung tissues were immediately collected into 1 mL cold PBS containing a protease and phosphatase inhibitor cocktail (Pierce, ThermoFisher Scientific) followed by homogenization by a dounce tissue homogenizer [6].

### 3.5. Hyperoxia-Induced HMGB1 Release in Macrophages

The effect of the composition in reducing the level of HMGB1 release was tested in murine immune cells in vitro. RAW264.7 cells either remained in room air (21% O_2_) or were exposed to 95% O_2_ for 24 h in the presence of the composition UP446 at concentrations of 0, 3.7, 11.1 and 33.3 μg/mL. HMGB1 levels in the cell culture media were determined by Western blot analysis described below in the assay section.

### 3.6. Phagocytosis Activity of Macrophages

The effect of the composition on the phagocytic activity of macrophages was tested in vitro. RAW264.7 cells either remained in room air (21% O_2_) or were exposed to 95% O_2_ for 24 h in the presence of a standardized composition (UP446) at concentrations of 3.7, 11.1, 33.3 and 100 μg/mL. Cells were then incubated with FITC-labeled latex mini-beads for one hour and stained with phalloidin and DAPI to visualize the actin cytoskeleton and nuclei, respectively. For quantification of phagocytic activity, at least 200 cells per group were counted, and the number of beads per cell was represented as a percentage of the 21% O_2_ (0 μg/mL) control group.

### 3.7. Assays

#### 3.7.1. Cytokines and Chemokines

The presence of TNF-α/IL-1β/IL-6 in undiluted rat serum was measured using the rat TNF-α/IL-1β/IL-6 Quantikine ELISA kit from R&D Systems (product#: RTA00, RLB00 and R6000B, respectively) as follows: undiluted serum was added to a microplate coated with TNF-α/IL-1β/IL-6 antibody. After 2 h at room temperature, TNF-α/IL-1β/IL-6 in serum was bound to the plate and the plate was thoroughly washed. Enzyme-conjugated TNF-α/IL-1β/IL-6 antibody was added to the plate and allowed to bind for 2 h at room temperature. The washing was repeated, and enzyme substrate was added to the plate. After developing for 30 min at room temperature, a stop solution was added, and the absorbance was read at 450 nm. The concentration of TNF-α/IL-1β/IL-6 was calculated based on the absorbance readings of a TNF-α/IL-1β/IL-6 standard curve.

The presence of CRP in rat bronchoalveolar lavage (BAL) diluted 1:1000 was measured using the C-Reactive Protein (PTX1) Rat ELISA kit from Abcam (product#: ab108827) as follows: 1:1000 diluted BAL was added to a microplate coated with CRP antibody. After 2 h on a plate shaker at room temperature, CRP in BAL was bound to the plate and the plate was thoroughly washed. Biotinylated C-Reactive Protein Antibody was added to the plate and allowed to bind for 1 h on a plate shaker at room temperature. The washing was repeated, and Streptavidin-Peroxidase Conjugate was added to the plate. After incubating for 30 min at room temperature, washing was repeated, and chromogen substrate was added. After developing for 10 min at room temperature, a stop solution was added, and the absorbance was read at 450 nm. The concentration of CRP was calculated based on the absorbance readings of a CRP standard curve.

Cytokine-induced neutrophil chemoattactant 3 (CINC-3)/macrophage inflammatory protein 2 (MIP-2) ELISA was carried out as follows: 50 µL of each rat lung homogenate sample (10 per group for vehicle, UP446 low dose, UP446 high dose, 7 per group for control) and 50 µL of assay diluent buffer was added to the wells of a 96-well microplate coated with monoclonal CINC-3 antibody and allowed to bind for 2 h. The plate was subjected to 5 washes before an enzyme-linked polyclonal CINC-3 antibody was added and allowed to bind for 2 h. The wells were washed another 5 times before a substrate solution was added to the wells and the enzymatic reaction was allowed to commence for 30 min at room temperature protected from light. The enzymatic reaction produced a blue dye that changed to yellow with the addition of the stop solution. The absorbance of each well was read at 450 nm (with a 580 nm correction) and compared to a standard curve of CINC-3 in order to approximate the amount of CINC-3 in each rat lung homogenate sample.

#### 3.7.2. Bacterial Counts

Bacterial counts in the lung homogenate and bronchoalveolar lavage were quantitated in serially diluted Luria–Bertani broth using a colony formation unit assay plated onto Pseudomonas Isolation Agar (Difco, Sparks, MD, USA) at 37 °C for 18 h.

#### 3.7.3. Western Blot Analysis

Cells were washed three times with PBS and lysed using a cell lysis buffer (Cell Signaling Technology, Danvers, MA, USA) supplemented with Halt Protease and Phosphatase Inhibitor Cocktail (78440, ThermoFischer, Waltham, MA, USA) for intracellular protein analysis. The total protein content of cell lysate was determined by using the Bicinchoninic Acid (BCA) assay kit (23225, ThermoFisher, Waltham, MA, USA), as per the manufacturer’s instructions. Samples were loaded onto 12% or 15% SDS-polyacrylamide gels (Bio-Rad, Hercules, CA, USA) and transferred to Immobilon-P membranes (Millipore, Bedford, MA, USA). Nonspecific binding sites on the membrane were blocked by incubating the membrane with 5% nonfat dry milk (Bio-Rad, Hercules, CA, USA) in Tris-buffered saline, containing 0.1% Tween 20 (TBST), for 1 h at room temperature. Next, the membranes were washed three times with TBST, and incubated overnight at 4 °C with anti-HO-1 (1:1000, #ab13248, Abcam, Cambridge, UK) and anti-pan-actin (1:1000, #8456, Cell Signaling) antibodies, diluted in 5% nonfat dry milk in TBST. After three washes with TBST, the membranes were incubated with goat anti-rabbit horseradish peroxidase-coupled secondary antibody (1:5000; GE Healthcare, Chicago, IL, USA) for 1 h at room temperature. Subsequently, membranes were again washed three times with TBST, and the immunoreactive proteins were visualized using the SuperSignal West Pico Plus Chemiluminescent Substrate (ThermoFisher, Waltham, MA, USA), as per the manufacturer’s instructions. Images were obtained using the Bio-Rad ChemiDoc XRS imaging system (Bio-Rad, Hercules, CA, USA). The immunoreactive bands were quantified using ImageJ software (version 2.0.0).

#### 3.7.4. Statistical Analysis

Data were analyzed using Sigmaplot (Version 11.0, San Jose, CA, USA). The results are represented as mean ± one SD. Statistical significance between groups was calculated by means of single factor analysis of variance followed by a paired *t*-test. *p*-values less than or equal to 0.05 (*p* ≤ 0.05) were considered statistically significant. When the normality test failed, for nonparametric analysis, data were subjected to Mann–Whitney sum ranks for *t*-test and Kruskal–Wallis one-way analysis of variance on ranks for ANOVA. When the treatment group sizes were unequal, Dunn’s test was used for pairwise comparisons and comparisons against the placebo group following rank-based ANOVA.

## 4. Discussion

Poor air quality from recent wildfires and other sources of air pollution poses a major public health threat worldwide. Antioxidants could potentially be considered as frontline defense and/or adjunct for air-pollution-induced oxidative stress damage of the lung. Historically, herbal medicines have been considered as an alternative to pharmaceuticals because of their relative safety; however, scientifically sound, research-backed, safe and efficacious nutritional supplements from natural sources for respiratory system support (i.e., mitigation of pollution-induced oxidative stress damage) are very limited. *Scutellaria baicalensis* and *Acacia catechu*, known antioxidants, have long been used in Traditional Chinese Medicine (TCM) and ayurvedic preparations, respectively, for centuries for varieties of human ailments. For instance, Radix Scutellaria has been reported as the second most utilized herb, with a 38% frequency in all TCM compositions for the treatment of respiratory tract infections [20]. Findings depicted in this report may support the frequent historical usage of these botanicals for respiratory system support.

Recent discoveries in pulmonary pathophysiology have shown extracellular HMGB1 as an alarmin to trigger profound inflammatory and immune responses following oxidative-stress-induced lung injury [11]. In the current report, we evaluated UP446 (an antioxidant composition) in multiple pre-clinical in vivo and in vitro models suggestive of its respiratory-system-support-related mode of action with the direct or indirect involvement of extracellular HMGB1.

The hypothesis that the composition could be involved in reducing the level of extracellular HMGB1 was first tested in a sepsis model indirectly. The composition UP446 effectively prevented the development of sepsis in a Lipopolysaccharide (LPS)-induced sepsis model. Mortality was significantly reduced (i.e., 50%) as a result of UP446 supplementation. Since mortality was the end point measurement, we did not measure the level of serum HMGB1 in this study; however, previously, it was reported that there is a correlation between high serum levels of HMGB1 and sepsis, resulting in an unbalanced host immune response, culminating in the death of the mice [33]. As such, the increased survival rate in UP446-treated animals in the current study could possibly be linked to the reduction in extracellular HMGB1 and mitigation of its downstream inflammatory effects.

We proceeded to assess the impact of the composition on acute inflammatory lung injury using an LPS-induced lung injury in rats. Intratracheal instillation of lipopolysaccharide (LPS) into the lungs of animals triggers an intense inflammatory response. It causes acute inflammatory lung injury that mimics the pathology in humans, characterized by pathological changes, such as diffuse alveolar damage, accompanied by profound increases in proinflammatory cytokines, infiltration of polymorphonuclear cells, increased alveolar capillary membrane permeability, and accumulation of protein-rich fluid in the alveolar space, leading to edema [34,35,36]. These pathological changes may be a direct consequence of increased accumulation of HMGB1 in the airways and the lungs, causing severe inflammatory injury. It has been reported that extracellular HMGB1 triggers the production of potent proinflammatory cytokines and chemokines, such as TNF-α, IL-1β, CINC-3, and IL-6, increased endothelial/epithelial barrier permeability, infiltration of polymorphonuclear cells, and formation of lung edema, causing lung injury and loss of lung function [12,37]. As such, major cytokines and chemotactic factors involved in acute inflammatory response in the lung have significant clinical relevance in cytokine storm intervention and alleviating the severity of acute respiratory distress syndrome (ARDS). In the current study, the composition UP446 mitigated LPS-induced acute inflammatory lung injury, possibly through inhibition of HMGB1 though our assay failed to show differences in the airway HMGB1 level in this model, suggesting the presence of additional pathways for the composition.

Nevertheless, as evidenced by the histopathology data, a statistically significant reduction in cellular and structural damage was observed as a result of oral supplementation of the composition UP446. Marked reductions in pulmonary edema further strengthened the lung protection capacity of the composition, possibly through a reduction in extracellular HMGB1 and hence improving the phagocytic activity of alveolar macrophages. Effective alleviation of acute inflammatory lung injury by the composition UP446 was further demonstrated by a reduction in key proinflammatory cytokines and chemokines (TNF-α, IL-1β and IL-6 and CINC-3) as well as the inflammatory protein CRP. Substantiating our findings, it was reported that the blend of *Scutellaria* and *Acacia* was found to decrease lung wet-to-dry weight ratio, mitigate lung histopathological changes, and reduce the release of inflammatory mediators, such as TNF-α and IL-1β, in BAL in a rat model of acute lung injury (ALI) [38]. Similarly, baicalein, a phenolic flavonoid from the root of *Scutellaria baicalensis* has been shown to significantly mitigate LPS-induced lung edema and attenuate the levels of IL-1β, TNF-α, IL-6, CINC-3 in broncho alveolar lavage fluid, with marked improvement of lung histopathological symptoms, in an LPS-induced rat model of acute inflammatory lung injury [39]. The major constituent of UP446, baicalin, has also been found to inhibit the cytoplasmic translocation of HMGB1 induced by lipopolysaccharide in vitro. In vivo, baicalin decreased serum HMGB1, TNF-α, IL-1β and IL-6 while improving survival and tissue injury of septic mice [40].

We further carried out an additional preclinical in vivo study to better understand the mode of action of the composition UP446 in a model that mimics mechanical ventilation-induced oxidative stress and secondary pulmonary bacterial infection. It has been previously shown that exposure to hyperoxia can compromise the host defense against bacterial infections, resulting in higher bacterial loads in lung tissues and airways and increased total protein content in lung lavage fluids upon microbial infection [6]. In the current study, we observed these phenomena in the vehicle-treated disease model with significant increases in bacterial load and protein exudate, whereas supplementation with UP446 showed significant reductions in both protein and bacterial load, suggesting the standardized botanical composition’s ability to improve host defense mechanisms, possibly through the reduction in extracellular HMGB1.

Accumulation of extracellular HMGB1 in the airways compromises innate immunity, leading to the impaired ability of alveolar macrophages to clear invading pathogens, exacerbating inflammatory lung injury. HMGB1-triggered cytokine storm can subsequently cause severe acute respiratory tract and lung injury that could ultimately cause death [6,38]. The involvement of HMGB1 in the pathogenesis of acute lung injury has been demonstrated in mouse models whereby the intratracheal administration of anti-HMGB1 antibodies mitigated the development of lung injury and increased survival rates [37]. To determine whether UP446-attenuated acute lung injury in mice exposed to hyperoxia and PA is due to its impact on the accumulation of extracellular HMGB1 in the airways, the levels of HMGB1 were measured in the lung lavage fluids. Prolonged exposure of the mice to hyperoxia and microbial infection increased the accumulation of HMGB1 in the airways by about 5-fold. Pretreating animals with UP446 showed a 71.6% reduction in the level of HMGB1 expression compared to vehicle-treated mice exposed to hyperoxia and PA infection. These data suggest that the composition UP446 can reduce the accumulation of airway HMGB1 in mice exposed to hyperoxia and PA infection. These findings suggest the enhanced ability of UP446 to improve host defense mechanisms against oxidative stress and microbial infection involving the respiratory system. Previously, the antioxidant activity of the composition through increasing endogenous antioxidant enzyme has been demonstrated in a human clinical study. In this study, healthy participants who were supplemented with the composition for 56 days orally showed a statistically significant increase in the serum level of glutathione peroxidase (GSH-Px) both at the interim (day 28) and the end of study (day 56) compared to baseline [20].

## 5. Conclusions

Reducing the level of extracellular HMGB1 induced by oxidative stress and/or respiratory tract infection could have a significant impact on protecting the lungs from injury and preserving normal physiological respiratory tract functions. The compilations of data in this report strongly suggest that the antioxidant composition UP446 could be indicated for lung protection through mitigation of HMGB1. Further confirmatory studies focused on the impact of the composition on this molecular target in humans is suggested.

## Figures and Tables

**Figure 1 molecules-28-06560-f001:**
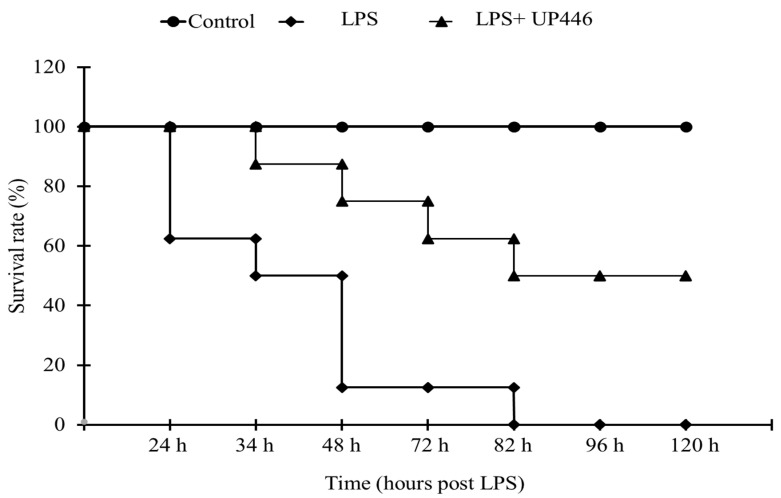
Effect of UP446 on the survival rate of mice with LPS-induced endotoxemia. Eight weeks old, male CD-1 mice (n = 8) were used in this study. While the control mice were injected with only PBS, the model mice (LPS) were pretreated with composition—UP446 for 7 days before lethal dose intraperitoneal injection of LPS at 25 mg/kg with a 10 mL/kg PBS volume an hour after the last dose. Animals were observed hourly and monitored for 5 days after LPS injection. The survival rate was calculated as: 100 − [(deceased mice/total number of mice) × 100]%.

**Figure 2 molecules-28-06560-f002:**
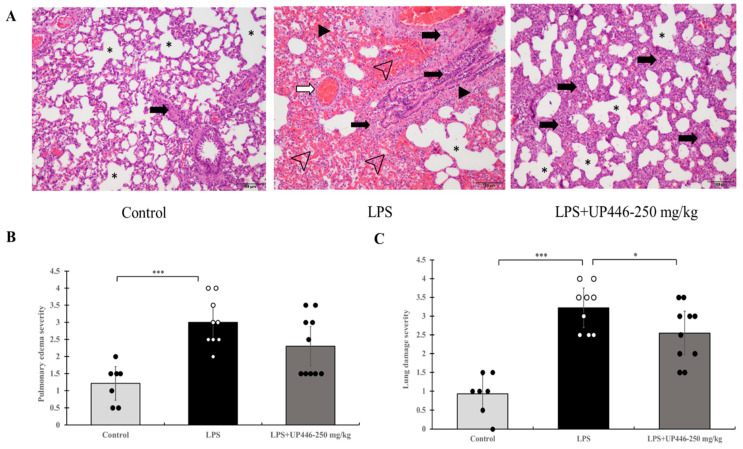
Effects of UP446 on lung histopathology from LPS-induced acute inflammatory lung injury model in rats. Male SD rats (n = 10) at the age of 9 weeks old were treated with the composition—UP446 orally for 7 days before the treatment with LPS. On the 8th day, an hour after oral treatment, LPS was instilled intratracheal (i.t.) at 10 mg/kg dissolved with PBS. The control rats (n = 7) received only PBS. Treatment groups include G1= control, G2 = LPS, G3 = LPS + UP446-125 mg/kg and G4 = LPS + UP446-250 mg/kg. G1 and G2 received the carrier vehicle (i.e., 0.5% CMC) during treatment period. G1, G2 and G4 were used for histopathological analysis. Rats were sacrificed 24 h post intratracheal LPS administration. At necropsy, the left lobe was dissected out, fixed in formalin and submitted to Nationwide Histology for histopathology analysis by a certified pathologist. Tissue and slide preparation were done per company protocol. (**A**) lung histopathology Magnification 100×. dark arrowhead: interstitial edema; open arrowhead: hemorrhagic alveolar sac; open arrow: endothelial damage and hemorrhage; dark arrow: infiltration of inflammatory cells mainly neutrophils and monocytes/macrophages; asterisk: alveolar space. (**B**) Pulmonary edema: alveolar, duct and bronchial, alveolar wall and Interstitial edema, congestion, hemorrhagic perivascular, alveolar sac edema, fibrin exudate, hemorrhagic alveolar sac, alveolar duct thicken dt Hyalin membrane type I loss, apoptotic cells, specific parameter scores 0–4. (**C**) Severity: Normal, minimum-mild, moderate, and extreme. Focal, regional, reginal extensive coalescing, diffuse, score 0–4. * *p* ≤ 0.05, *** *p* ≤ 0.00001.

**Figure 3 molecules-28-06560-f003:**
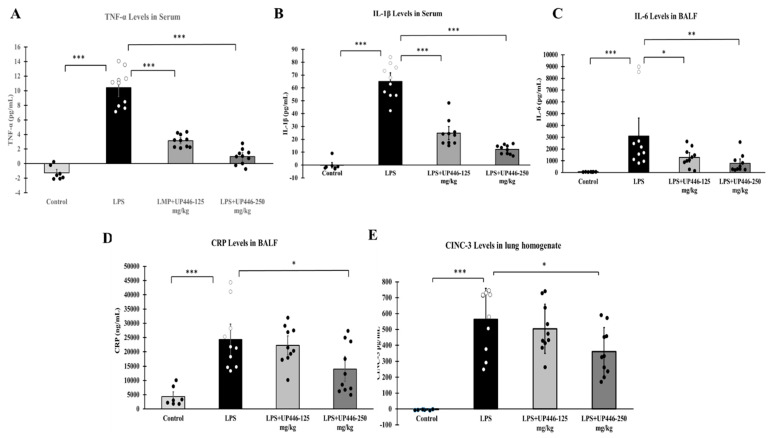
Effect of UP446 on inflammatory cytokines, and chemokines in LPS-induced ALI in rats. Male SD rats (n = 10) at the age of 9 weeks old were treated with the composition—UP446 at 125 mg/kg and 250 mg/kg orally for 7 days before the treatmpleent with LPS. On the 8th day, an hour after oral treatment, LPS was instilled intratracheal (i.t.) at 10 mg/kg dissolved with PBS. The control rats (n = 7) received only PBS. Treatment groups include G1 = control, G2 = LPS, G3 = LPS + UP446-125 mg/kg and G4 = LPS + UP446-250 mg/kg. G1 and G2 received the carrier vehicle (i.e., 0.5% CMC) during treatment period. Rats were sacrificed, serum for TNF-α (**A**) and IL-1β (**B**) and bronchoalveolar lavage (BAL) for IL-6 (**C**) and CRP (**D**) was collected 24 h post intratracheal LPS administration. The right lobe was homogenized for MIP-2/CINC-3 (**E**) activity analysis. * *p* ≤ 0.05; ** *p* ≤ 0.001; *** *p* ≤ 0.00001.

**Figure 4 molecules-28-06560-f004:**
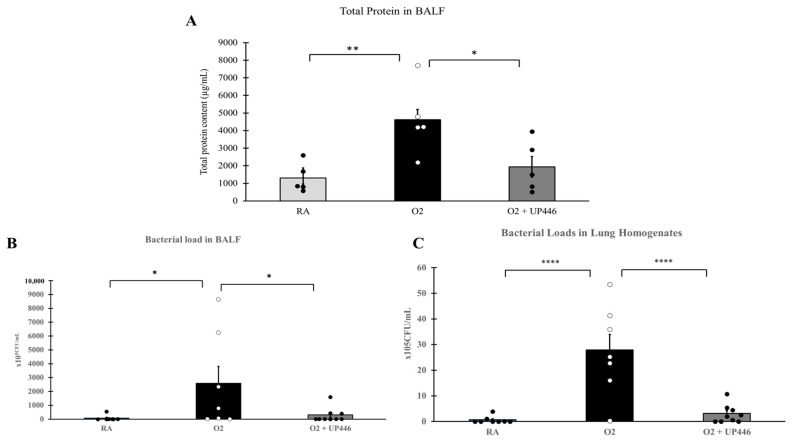
Effect of UP446 on BALF total protein (**A**), BALF bacterial load (**B**) and lung homogenate bacterial load (**C**) in oxidative stress/bacterial challenged mice. Following treatment with the composition UP446 (250 mg/kg) orally for seven days, mice were exposed to >90% oxygen for 48 h and continued oral treatment for 2 more days before being inoculated with *Pseudomonas aeruginosa* (PA). Mice were euthanized 24 h after bacteria inoculation; lungs were lavaged, and the lavage fluid was used to determine the total protein content (**A**) and bacterial load (**B**). Bacterial load was also determined from the lung homogenate (**C**). * *p* ≤ 0.05, ** *p* ≤ 0.001; **** *p* ≤ 0.0001.

**Figure 5 molecules-28-06560-f005:**
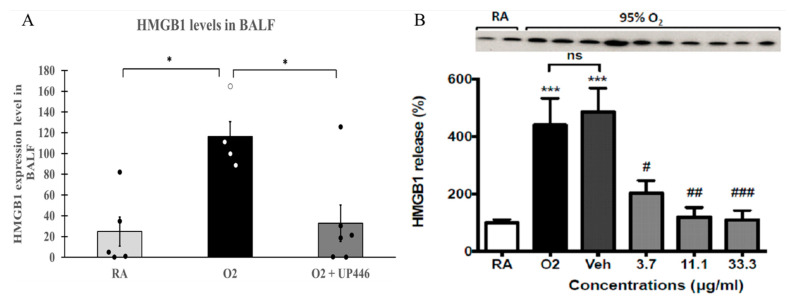
(**A**) Effect of UP446 on HMGB1 in oxidative stress/bacterial challenged mice. Following treatment with the composition UP446 (250 mg/kg) orally for seven days, mice were exposed to >90% oxygen for 48 h and continued oral treatment for 2 more days before being inoculated with *Pseudomonas aeruginosa* (PA). Mice were euthanized 24 h after bacteria inoculation; lungs were lavaged, and the lavage fluid was used to determine HMGB1 expression levels. (**B**) Hyperoxia-induced HMGB1 release in RAW 264.7 cells: RAW 264.7 cells either remained at room air (21% O_2_) or were exposed to 95% O_2_ for 24 h in the presence of a composition at indicated concentrations. HMGB1 levels in the media were determined by Western blot analysis. Each value represents the mean ± SEM of 2 independent experiments, in duplicates. “ns”: no significance. * *p* ≤ 0.05, *** *p* < 0.001 compared to room air control (AR). # *p* < 0.05, ## *p* < 0.01, ### *p* < 0.001 compared to vehicle control.

**Figure 6 molecules-28-06560-f006:**
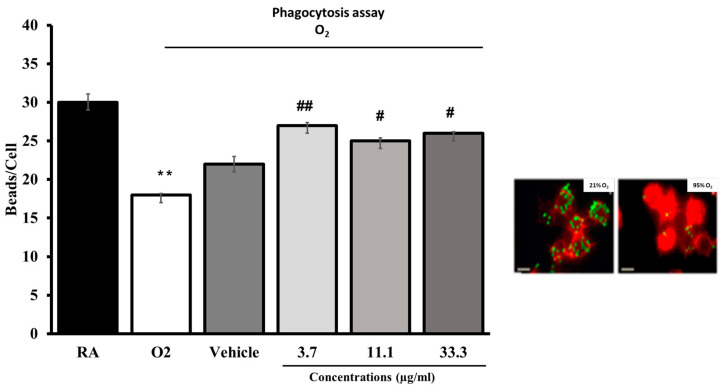
Hyperoxia-compromised macrophage phagocytic function. RAW 264.7 cells either remained at room air (21% O_2_) or were exposed to 95% O_2_ for 24 h in the presence of a standardized composition (specifics and conc.). Cells were then incubated with FITC-labeled latex mini-beads for 1 h and stained with phalloidin and DAPI to visualize the actin cytoskeleton and nuclei, respectively. For quantification of phagocytic activity, at least 200 cells per group were counted and the number of beads per cell was represented as a percentage of the 21% O_2_ (0 μg/mL) control group. Each value represents the mean ± SEM of 2 independent experiments for each group, in duplicates. Significance is compared to the 95% O_2_ (0 μg/mL) control group. ** *p* < 0.001 compared to room air control (AR). # *p* < 0.05, ## *p* < 0.01compared to vehicle control.

**Figure 7 molecules-28-06560-f007:**
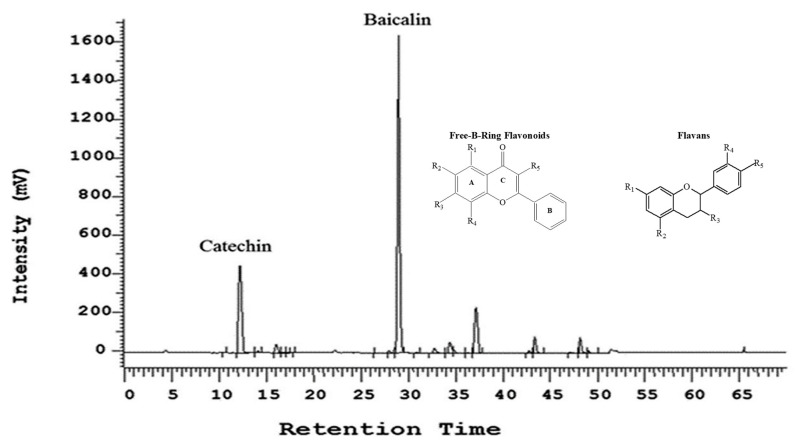
HPLC chromatogram and chemical structure of free B-ring flavonoid (baicalin) and flavan (catechin). The flavonoids were detected using a UV detector at 275 nm and identified based on retention time by comparison with known flavonoid standards. Free B-Ring Flavonoids: Baicalin: R1 = R2 = OH, R3 = Glucuronide, R4 = R5 = H; Baicalein: R1 = R2 = R3 = OH, R4 = R5 = H; Oroxylin A: R1 = R3 = OH, R2 = OMe, R4 = R5 = H; Chrysin: R1 = R3 = OH, R2 = R4 = R5 = H; Wogonin: R1 = R3 = OH, R4 = OMe, R3 = R5 = H; Wogonin 7-Glucuronide: R1 = OH, R3 = Glucuronide, R4 = OMe, R2 = R5 = H. Flavans: Catechin (+): R1 = R2 = R3 = R4 = R5 = OH; Epicatechin (−): R1 = R2 = R3 = R4 = R5 = OH.

## Data Availability

Conclusions were made based on data depicted in this report. Raw data could be requested through corresponding authors.

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
