# Peer review of "A Standardized Botanical Composition Mitigated Acute Inflammatory Lung Injury and Reduced Mortality through Extracellular HMGB1 Reduction"

_molecules, 2023, doi:10.3390/molecules28186560_

Round 1

Reviewer 1 Report

This study by Yimam et al investigated about UP446 obtained from two plants and evaluated its efficacy on attenuating acute inflammatory lung injury, sepsis. This study used three models included murine acute lung injury, LPS-induced sepsis, and LPS-induced acute inflammatory lung injury to assess the protective effect. They found that the phagocytic activity of UP446 and HMGB1 release in macrophages in vitro. They also performed HE stain for lung tissue, ELISA for inflammatory cytokines and chemokines, western blot analysis for protein and HMGB1. The results showed that UP446 significant decrease the mortality, proinflammatory cytokines in serum and bronchoalveolar lavage, increased bacterial clearance of airways and lungs. However, authors need to revise some major points, as listed below, to make this study complete.

Major

1.     Authors demonstrated some advantages of baicalin in introduction section, why don’t you just use baicalin instead of UP446? Are there any reasons? I suggest authors could replenish a paragraph about this issue.

2.     In Figure 5, the western blot analysis should provide internal control such as b-actin and the name of group should also be provided under the lane.

3.     Except for HMGB1, authors should discuss some other protein associated with your hypothesis such as NF-kB, TLR etc in one of model.

4.     Authors mentioned regulation of phagocytosis was involved in the MOA of UP446, I suggested authors try to express the UP446 affect phagocytosis via HMGB1 and further inhibit downstream inflammatory pathway. However, authors should replenish more molecular evidence to support the hypothesis.

Minor

1.     Authors should supply a graphic abstract to help us to understand the whole picture of this research.

2.     In line 500 “Nevertheless, ss evidenced by the histopathology data” must be wrong, please revise it.

Moderate editing of English language required

Author Response

Date: September 1, 2023

Dear Editor,

We express our utmost appreciation and gratitude to the editorial office and reviewers for their time and invaluable constructive comments. We have revised the full manuscript according to the suggestions and incorporated our response at appropriate sections of the body and figures as follows:

Reviewer 1:

Major

  1. Authors demonstrated some advantages of baicalin in introduction section, why don’t you just use baicalin instead of UP446? Are there any reasons? I suggest authors could replenish a paragraph about this issue.

Reply: Baicalin backgrounds were provided in the introduction section as it is the major component of UP446. Composition details have been provided in the materials and method section.

A paragraph has been added to expand the introduction of UP446: “UP446 is a standardized composition consisting primarily of free B-ring flavonoid, baicalin, from S. baicalensis and a flavan, catechin, from the heartwoods of A. catechu as detailed in the materials and method section. The composition is a dual cyclooxygenase (COX) and lipoxygenase (LOX) enzymes inhibitor which was found to decrease mRNA expression and protein levels of the proinflammatory cytokines, such as interleukin (IL)-1β, IL-6, and tumor necrosis factor (TNF)-α in preclinical studies (Burnett et al., 2007; Tseng-Crank et al. 2010). Recently, it has been reported that, the composition was considered beneficial in mounting a robust humoral response (elevated total IgA and IgG levels) following influenza vaccination paired with strong antioxidation capacity (increased glutathione peroxidase) in healthy participants (Lewis et al., 2023). These preclinical and clinically proven anti-inflammatory, antioxidant and immune support properties of the composition may have significant contribution for a healthy respiratory system. The current studies were designed to explore additional mechanisms by which the composition could provide protection to the lung and/or the respiratory system in general.”

  1. In Figure 5, the western blot analysis should provide internal control such as b-actin and the name of group should also be provided under the lane.

Reply: Data have been represented relative to the room air (RA). The bar graphs were plotted to align the duplicates for each treatment group described in the X-axis.

  1. Except for HMGB1, authors should discuss some other protein associated with your hypothesis such as NF-kB, TLR etc in one of model.

Reply: There are future plans to run additional assays to assess the impact of the composition on other proteins involved in the inflammatory pathways.

  1. Authors mentioned regulation of phagocytosis was involved in the MOA of UP446, I suggested authors try to express the UP446 affect phagocytosis via HMGB1 and further inhibit downstream inflammatory pathway. However, authors should replenish more molecular evidence to support the hypothesis.

 Reply: Thank you for the suggestion. We do have future plans to carry out more assays to explore the indicated pathways.

Minor

  1. Authors should supply a graphic abstract to help us to understand the whole picture of this research.

Reply:  The following has been inserted as a graphic abstract. Please see the attached.

  1. In line 500 “Nevertheless, ss evidenced by the histopathology data” must be wrong, please revise it.

Reply: Revised for correction

Reviewer 2 Report

The authors present a manuscript entitled “A standardized bioflavonoid composition mitigated acute inflammatory lung injury and reduced mortality through HMGB1 reduction”. The idea of the article was that a standardized bioflavonoid antioxidant composed from two plants: Scutella baicalensis and Acacia catechu was evaluated for its efficacy in attenuating acute inflammatory lung injury and sepsis induced by oxidative stress and endotoxemia. The authors examined the pharmacology effect based on in vitro and in vivo models which are in good agreement between in vitro and in vivo assays. The bioflavonoid itself contained major flavonoid: baicalin (60%), cathecin (10%) and minor flavonoids: wogonin 7-glucuronide and baicalein. I suggest to draw the structure the composition of flavonoid at least the major one. The pharmacology effect of the single major component should be discussed and compared with current formulation. Moreover, only one formulation (UP446) evaluated both in vitro and in vivo is difficult to understand as the authors claimed for bioflavonoind compositions. Is this research to characterize the pharmacology effect only for UP446? If so, the title should be changed and all data formulation related to the disease should be described. The authors should introduce or explain why they choose only specific composition of UP446 for the only material for examination. Why don’t the authors check other formulations for the same pharmacological effect and compared with UP446? In short, using both in vitro and in vivo models for assays for justification of the results are good, but using only one formulation (UP466) without any detail information and justification is insufficiency.

The English is fine, but need to check more detail for grammatical errors.

Author Response

  1. I suggest to draw the structure the composition of flavonoid at least the major one.

Reply: The manuscript has been revised to include the structure and chromatogram of the two main component of the composition: baicalin and catechin.

(attached)

Figure 1: HPLC chromatogram and chemical structure of free-B-ring flavonoid (baicalin) and flavan (catechin). The flavonoids were detected using a UV detector at 275 nm and identified based on retention time by comparison with known flavonoid standards

  1. The pharmacology effect of the single major component should be discussed and compared with current formulation.

Reply: Relevant pharmacological effects in regard to the anti-inflammatory, antioxidant, immune and respiratory system support of the major components of the composition have been discussed both in the introduction and discussion section of the manuscript; however, direct comparison of the main constituents (baicalin and catechin) against the composition (UP446) has not been done as it was beyond the scope of this report.  We plan to carry out comparative study in the future.

  1. Moreover, only one formulation (UP446) evaluated both in vitroand in vivo is difficult to understand as the authors claimed for bioflavonoind compositions. Is this research to characterize the pharmacology effect only for UP446? If so, the title should be changed and all data formulation related to the disease should be described.

Reply: Revised the manuscript through out to remove the word “bioflavonoid” for clarity. The title has been revised to “A standardized botanical composition mitigated acute inflammatory lung injury and reduced mortality through HMGB1 reduction.”

  1. The authors should introduce or explain why they choose only specific composition of UP446 for the only material for examination. Why don’t the authors check other formulations for the same pharmacological effect and compared with UP446? In short, using both in vitro and in vivo models for assays for justification of the results are good, but using only one formulation (UP466) without any detail information and justification is insufficiency.

Reply: The scope of the manuscript was to evaluate the pharmacological effect of UP446 in vivo and in vitro using the disclosed models. As suggested, we have inserted the following paragraph to expand the introduction of UP446. “ UP446 is a standardized composition consisting primarily of free B-ring flavonoid, baicalin, from S. baicalensis and a flavan, catechin, from the heartwoods of A. catechu as detailed in the materials and method section. The composition is a dual cyclooxygenase (COX) and lipoxygenase (LOX) enzymes inhibitor which was found to decrease mRNA expression and protein levels of the proinflammatory cytokines, such as interleukin (IL)-1β, IL-6, and tumor necrosis factor (TNF)-α in preclinical studies (Burnett et al., 2007; Tseng-Crank et al. 2010). Recently, it has been reported that, the composition was considered beneficial in mounting a robust humoral response (elevated total IgA and IgG levels) following influenza vaccination paired with strong antioxidation capacity (increased glutathione peroxidase) in healthy participants (Lewis et al., 2023). These preclinical and clinically proven anti-inflammatory, antioxidant and immune support properties of the composition may have significant contribution for a healthy respiratory system. The current studies were designed to explore additional mechanisms by which the composition could provide protection to the lung and/or the respiratory system in general. 

Round 2

Reviewer 1 Report

The correction of the manuscript is just fine.

Author Response

Date: September 4, 2023

Dear Editor,

We sincerely appreciate the editorial office and reviewers for their quick response. Because of your feedback the current version of the manuscript is in a much better standing in its content and appearance from its original version. 

Thank you!

Reviewer 2 Report

Thank you for sending the revision. For final suggestion, please draw the chemical structure in Figure 2 using an appropriate software such as chemdraw. The name of compounds in Fig 2. is not so clearly. Please increase the size of fonts in Figure 2. After extensive correction, I recommend to accept this manuscript.

Please check again for grammatical errors.

Author Response

Date: September 4, 2023

Dear Editor,

We sincerely appreciate the editorial office and reviewers for their quick response. Because of your feedbacks the current version of the manuscript is in a much better standing in its content and appearance from its original version. 

We have revised the full manuscript as suggested:

Reviewer 2:

For final suggestion, please draw the chemical structure in Figure 2 using an appropriate software such as chemdraw. The name of compounds in Fig 2. is not so clearly. Please increase the size of fonts in Figure 2.

Reply: We have revised the manuscript as suggested. (please see the attached)
